# In Vivo Evaluation of *Dendropanax morbifera* Leaf Extract for Anti-Obesity and Cholesterol-Lowering Activity in Mice

**DOI:** 10.3390/nu13051424

**Published:** 2021-04-23

**Authors:** Ji-Hye Song, Hyunhee Kim, Minseok Jeong, Min Jung Kong, Hyo-Kyoung Choi, Woojin Jun, Yongjae Kim, Kyung-Chul Choi

**Affiliations:** 1Department of Biomedical Sciences, Asan Medical Center, University of Ulsan College of Medicine, Seoul 05505, Korea; EIU1258162423710@gmail.com (J.-H.S.); ttlok1816@naver.com (H.K.); alstjr9291@naver.com (M.J.); beebee1125@naver.com (M.J.K.); 2Korea Food Research Institute, Jeollabuk-do 55365, Korea; chkyoung@kfri.re.kr; 3Division of Food and Nutrition, Chonnam National University, Gwangju 61186, Korea; wjjun@chonnam.ac.kr; 4SDC Research Institute, Jeollanam-do 57309, Korea; yjds06@nate.com

**Keywords:** *Dendropanax morbifera*, methylisobutylxanthine, in vivo, obesity, cholesterol

## Abstract

Metabolic syndrome is a worldwide health problem, and obesity is closely related to type 2 diabetes, cardiovascular disease, hypertension, and cancer. According to WHO in 2018, the prevalence of obesity in 2016 tripled compared to 1975. *D. morbifera* reduces bad cholesterol and triglycerides levels in the blood and provides various antioxidant nutrients and germicidal sub-stances, as well as selenium, which helps to remove active oxygen. Moreover, *D. morbifera* is useful for treating cardiovascular diseases, hypertension, hyperlipidemia, and diabetes. Therefore, we study in vivo efficacy of *D. morbifera* to investigate the prevention effect of obesity and cholesterol. The weight and body fat were effectively reduced by *D. morbifera* water (DLW) extract administration to high-fat diet-fed C57BL/6 mice compared to those of control mice. The group treated with DLW 500 mg∙kg^−1^∙d^−1^ had significantly lower body weights compared to the control group. In addition, High-density lipoprotein (HDL) cholesterol increased in the group treated with DLW 500 mg∙kg^−1^∙d^−1^. The effect of DLW on the serum lipid profile could be helpful to prevent obesity. DLW suppresses lipid formation in adipocytes and decreases body fat. In conclusion, DLW can be applied to develop anti-obesity functional foods and other products to reduce body fat.

## 1. Introduction

Obesity is a worldwide health problem and an indicator of metabolic syndrome; it is closely related to type 2 diabetes, cardiovascular disease, hypertension, and cancer. According to WHO in 2018, the prevalence of obesity in 2016 tripled compared to 1975. In the United States, 68% of the adult population is overweight, and more than half have a health risk. To reduce the prevalence and to effectively treat obesity, it is necessary to study its regulatory mechanisms. Fat cells store energy in our bodies and secrete a variety of cytokines that regulate signal transduction in adipose tissue and muscle [1]. Differentiation of adipocytes is associated with changes in cell morphology, hormone sensitivity, and gene expression. Dexamethasone (DEX) and isobutyl-methylxanthine (IBMX) can promote CCAAT-enhancer-binding proteins beta (C/EBPβ) and gamma (C/EBPγ) at the early stage of adipogenesis, respectively. Then, both of them transactivate PPARγ and C/EBPα [2]. Expression of these transcription factors increases the expression of the peroxisome proliferator-activated receptor γ (PPARγ) transcription factor, which increases the expression of C/EBPα and promotes the expression of adipocyte differentiation-related genes in the later stages of differentiation. The target genes for PPARγ include those for adipocyte protein 2 and leptin, which are adipocyte-specific. Sterol regulatory element binding protein 1c (SREBP-1c) is mainly involved in fatty acid metabolism and lipid biosynthesis. Its expression is induced from the early stage of adipocyte differentiation, and it promotes differentiation and increases expression of fatty acid metabolism-related genes [3].

*Dendropanax morbifera* is known as a highly physiologically active material. Most of the *D. morbifera* have good ingredients in the sap. Since the amount of sap from one tree is small, and only from trees that are 10 to 15 years old or older, we used a 15-year-old *D. morbifera* and verified the efficacy of the leaves. *D. morbifera* was provided by HBJ Bio Farm, Jeju, Korea. *D. morbifera* leaf water extracts were extracted from the leaves of 15 years old trees grown, and the extraction process was described in the method. As a result of analyzing the leaf extract, quercetin and catechins (e.g., catechin, EGCG, EC, and ECG) were detected. In addition, various nutrients such as dietary fiber were also detected. Quercetin is a derivative of flavonol and is a yellow pigment contained in many plants. It is contained in large amounts in onion skins and is also used as a food additive or antioxidant. Quercetin is an ingredient that has a good effect on anticancer and blood vessel strengthening. Catechin is a type of polyphenol compound that gives off the astringent taste of green tea. It is also used as a vegetable antibacterial agent and is contained in green tea, black tea, oolong tea, and white tea. The action of catechins is known to have a good effect on lowering blood sugar, lowering cholesterol levels, removing free radicals, and preventing aging. Therefore, to verify the other efficacy of *D. morbifera* leaf extract, we additionally conducted a study on anti-obesity and cholesterol-lowering. *D. morbifera* in the Araliaceae family is a well-known wild ginseng plant [4]. It is called the ginseng tree because it is rich in saponin and contains a very rich tannin component [5], which is known to release cholesterol from the body. *D. morbifera* reduces the levels of bad cholesterol and triglycerides in the blood and provides a variety of antioxidant nutrients and germicidal substances, as well as selenium, which aids in the removal of active oxygen [6]. Therefore, *D. morbifera* is useful for treating cardiovascular diseases, hypertension, hyperlipidemia, and diabetes [7]. *D. morbifera* produces a yellowish resinous liquid when the bark of the tree is scratched [8]. It is a high-grade paint with a sedative effect because it contains benzoic acid. In the literature of Samguk Sagi, it was used to coat the emperor’s armor, helmet, and accessories. The roots and stems of the D. morbifera tree have a sweet taste, warm quality and are effective for a decreased risk of a stroke and stimulates blood circulation. Antioxidants make up 85% and 75% of the extracts of the *D. morbifera* leaf and stem, respectively. In addition, these substances have growth-inhibitory effects in chloroquine-sensitive strains and enhance immune cell production, thereby protecting against various diseases by enhancing the early immune system and other defense systems. The hydrothermal extracts of the *D. morbifera* trees inhibit oxidative damage caused [6] by alcohol in the HepG2/2E1 cell line and increase the activity of antioxidant enzymes in experimental animals to effectively remove free radicals. In addition, oleifoliosides A, B, and cycloartane-type glycosides extracted from the *D. morbifera* tree inhibit the secretion of various inflammatory mediators [9] and promote the expression of the Apoptogenic Factors (AIF) and EndoG proteins [10], which are apoptosis inducers in cancer cells. Conversely, the essential oil component of the *D. morbifera* tree induces a beneficial effect on lipids by decreasing total cholesterol (TC), LDL cholesterol, and triglyceride (TG) and increasing HDL cholesterol in mice [11]. Although there are some studies on the efficacy of *D. morbifera*, research on the therapeutic efficacy and mechanism of action is limited, and there is little research on the possibility of practical use as food [7].

Previous studies have shown that DLW has an anti-obesity effect and lowers cholesterol in 3T3-L1 cells. Therefore, this study investigated the anti-obesity efficacy and mechanism of action in vivo. Before entering the in vivo experiment, toxicity studies were conducted because there were insufficient data on toxicity studies for *D. morbifera*. We compared the levels according to changes in external changes such as body weight and fat weight, and cholesterol such as TC and HDL using mice induced with a high-fat diet. In this experiment, anti-obesity and cholesterol-lowering effects were investigated. Further-more, the expression of obesity-related factors was measured to investigate the mechanisms. This study assessed the efficacy of *D. morbifera* leaf water (DLW) on obesity.

## 2. Materials and Methods

### 2.1. Preparation of D. morbifera Leaf Extracts

We used a 15-year-old *D. morbifera* and verified the efficacy of the leaves and were provided by HBJ Bio Farm, Jeju, Korea. *D. morbifera* leaf extracts were prepared by adding leaves 30 kg to 600 L volumes of water and incubating at 80 °C for 3 h. Subsequently, the extracts were concentrated to 60 L before freeze-drying, then dried extracts 4.5 kg were mixed and dissolved in water for 10 min. The solutions were then centrifuged at 24,300 g for 10 min and the supernatants were collected for the experiments. Mouse dose concentration calculation; in vitro dose in mg/ml * differences between species (10) * Km Mouse (3), (ex; 500 ug/mL = 15,000 ug/30 g (mouse average weight) = 0.5 mg/1 g = 500 mg/1000 g = 500 mg/kg).

### 2.2. Animal Experiment

Animal protocols were approved by the Institutional Animal Care and Use Review Committee of the University of Ulsan College (2017-02-148; Seoul, Korea) and conducted in accordance with relevant national legislation.

Short- or long-term toxicity experiment; After one week acclimation period, 3-week-old C57BL/6 mice, male, were fed a normal food and distributed into five groups (Con, Dex100, GAR400, DLW250, DLW500) of six mice each. The extracts used in the five groups were weighed after freeze-drying, dissolved in a certain amount of water, and injected orally; the mice were fed a standard diet (Control), Dextrin 100 mg (*w/w*; Dex100), Garcinia 400 mg (*w/w*; GAR400), containing 250 mg *D. morbifera* leaf water extracts (*w/w*; DLW250) or 500 mg *D. morbifera* leaf water extracts (*w/w*; DLW500).

High-fat induced experiment; 3-week-old C57/BL6 mice, male, were divided into six groups (6 mouse/group; Con, HF, HF+Dex100, HF+GAR400, HF+DLW250, HF+DLW500) and housed in a temperature and humidity-controlled room with a 12-h light/dark cycle with free access to food and water. After a one-week acclimation period, the mice were fed a standard diet (Control), 60% HFD (HF), 60% HFD+Dextrin 100 mg (*w/w*; HF+Dex100), 60% HFD+Garcinia 400 mg (*w/w*; HF+GAR400), 60% HFD+containing 250 mg *D. morbifera* leaf water extracts (*w/w*; HF+DLW250) or 500 mg *D. morbifera* leaf water extracts (*w/w*; HF+DLW500). Mice fed a standard diet (Control) without DLW were used as controls. Both the Control (98121701) and HF (D12492) were purchased from Research Diets, Inc. (New Brunswick, NJ, USA), then mixed with DLW at appropriate concentrations, and provided ad libitum.

Body weights were measured at the beginning of the experiment and subsequently at 1-week intervals for every day (short-term treatment in toxicity experiment) or 2 days (long-term treatment in toxicity experiment or high-fat induced experiment).

Similarly, the food consumption of each group was recorded at weekly intervals for every day (short-term treatment in toxicity experiment) or 2 days (long-term treatment in toxicity experiment or high-fat induced experiment).

The highest extract dosage used was 500 mg/kg body weight; control groups were given 250 or 500 mg/kg body weight of the vehicle.

The mice received oral doses once a day for a total of 8 days (short-term treatment in toxicity experiment), once every 2 days for a total of 30 days (long-term treatment in toxicity experiment), and once every 2 days for a total of 12 weeks (high-fat induced experiment).

### 2.3. Histological Examination

Mouse liver tissues were collected, fixed in 4% formalin for 24 h and then embedded in paraffin. Tissue sections of 4 μm in thickness were obtained using a microtome and stained with hematoxylin and eosin. Images of the tissue sections were taken using a light microscope at different magnifications. Liver tissue and white adipose tissue (WAT) samples were fixed in 4% buffered formalin and cut into 4 μm-thick sections. The sections were stained with hematoxylin and eosin and examined by light microscopy (OLYMPUS CKX41, Tokyo, Japan).

### 2.4. Plasma Biochemistry

After withdrawing blood from mice, plasma parameters were assessed, included alanine aminotransferase (ALT), aspartate aminotransferase (AST), creatine, and urea. Plasma was obtained by centrifuging the anticoagulated blood. The levels of TC and HDL cholesterol were analyzed using cholesterol assay kits (Asan Pharmaceutical, Seoul, Korea) according to the manufacturer’s instructions.

### 2.5. Determination of SOD Activity

The quantification of superoxide dismutase (SOD) activity in plasma was determined using a SOD determination kit (#S311; Dojindo Laboratories, Kumamoto, Japan) according to the manufacturer’s protocol. This method is based on the xanthine oxidase reaction, which induces superoxide production. In this assay, SOD activity is expressed as the ratio of decreased WST-1 formazan and was performed at 450 nm absorbance using a microplate reader (Model 550; Bio-Rad Laboratories, Hercules, CA, USA).

### 2.6. Determination of Lipid Peroxidation

The levels of lipid peroxidation products were evaluated using a lipid peroxidation (malondialdehyde, MDA) assay kit (ab118970, Abcam, Cambridge, UK). The liver tissue (10 mg) was homogenized according to the manufacturer’s protocol, with MDA-thiobarbituric acid (TBA) adducts formed by the addition of TBA solution. The TBA-MDA adducts were quantified colorimetrically by measuring the optical density at 532 nm using a microplate reader (Model 550; Bio-Rad Laboratories, Hercules, CA, USA).

### 2.7. Measurement of Nitric Oxide Activity

In vivo urine nitric oxide levels were examined by nitrite/nitrate quantitation using the Nitric Oxide Colorimetric Assay Kit (Cat#K262; Biovision; Milpitas, CA, USA) according to the manufacturer’s protocol. After the col-or was developed for 10 min at room temperature, the absorbance was measured at 540 nm in a plate reader using a microplate reader (Model 550; Bio-Rad Laboratories, Hercules, CA, USA).

### 2.8. Limulus Amebocyte Lysate Assays

Plasma endotoxin contents were determined using the Diazo-coupled limulus amebocyte lysate (LAL) assay (Associates of Cape Cod, Inc., E. Falmouth, MA, USA) according to the manufacturer’s protocol. The assay reagents-treated samples lead to the formation of a magenta derivative that absorbs light at 545 nm.

### 2.9. RNA Extraction and Real-Time Quantitative PCR Analyses

Epididymal fat from white adipose tissues obtained at the end of the animal experiments were washed with ice-cold PBS and lysed using RNA EasySpin kits (Intron Biotechnology, Seongnam, Korea) according to the manufacturer’s instructions. Briefly, tissues were weighed and cut into small pieces. To obtain the total RNA isolation, tissue pieces were homogenized in easy-blue buffer (Intron Biotechnology, Seongnam, Korea) using a handheld homogenizer (MIULAB, Zhejiang, China).

The total RNA from each sample was reverse transcribed using PrimeScriptTM Reverse Transcriptase (Takara Bio Inc., Shiga, Japan) according to the manufacturer protocols. All samples were normalized to mouse actin and were expressed as fold induction, and all reactions were performed in triplicate. Relative expression levels were calculated with SD using the comparative quantification method. The following primers were used: 5′-GACAGACTGATCGCAGAGAAAG-3′ and 5′-TGGAGAGCCCCACACACA-3′, ACC-1; 5′-ACCCTGAGGCATCTATTGACA-3′ and 5′-TGACATACTCCCACAGATGGC-3′, CPT1α; 5′-CCCATGTGCTCCTACCAGAT-3′ and 5′-CCTTGAAGAAGCGACCTTTG-3′, CPT1β; 5′-CATTCTCAGGCGAGAGTGACAT-3′ and 5′-GACGCGAAGCTCGTGGAT-3′, adipose triglyceride lipase (ATGL); 5′-TCCGCCTCTGGGCATTC-3′ and 5′-GAATCGGCCCACAATCCA-3′, DGAT1; and 5′-CTATGAGCTGCCTGACCGTC-3′ and 5′-AGTTTCATGGATGCCACAGG-3′, mACTIN.

### 2.10. Western Blot Analysis

Before Western blotting analysis, white adipose tissues obtained at the end of the animal experiments were washed with ice-cold PBS. WAT were then prepared, and tissue pieces were homogenized using a hand held homogenizer (MIULAB, Zhejiang, China) in lysis buffer containing 50-mM Tris-HCl (pH 7.5), 150-mM NaCl, 1% NP-40, 10-mM NaF, 10-mM sodium pyrophosphate, and protease inhibitors and were incubated for 30 min on ice. The lysates were then centrifuged at 20,000× *g* for 20 min at 4 °C, and total proteins were separated using 10% sodium dodecyl sulfate/polyacrylamide gel electrophoresis (SDS-PAGE) and were transferred to nitrocellulose membranes. Membranes were then blocked by incubating for 2 h in 5% (*w/w*) Bovine Serum Albumin (Cat.BSA0.1, Bovogen Biologicals Pty Ltd., Williams Avenue, East Keilor, VIC, Australia) in 1× phosphate buffered saline (PBS) containing Tween-20 (PBST). Blocked membranes were then incubated with primary antibodies for 2 h or overnight at 4 °C. We used the following antibodies: anti-Phospho-AMP-activated protein kinase α (pAMPKα; Cell Signaling Technology, #2531), anti- AMP-activated protein kinase α (AMPKα; Cell signaling Technology, #5831), and anti-b-actin (Sigma Aldrich, St. Louis, MO, USA, A5441). After washing three times with 1× phosphate buffered saline (PBS) containing Tween-20 (PBST), membranes were incubated with anti-mouse secondary horseradish peroxidase-conjugated antibody (Thermo Scientific, Rockford, IL, USA) for 2 h and visualized using Fusion Solo 6 S (Vilber Lourmat, Collegien, France) with an enhanced WesternBright ECL chemiluminescence detection reagent (Advansta).

### 2.11. Statistical Analysis

Statistical analyses were performed using GraphPad Prism 8.0.1 to calculate the mean and standard deviation. The difference in mean between each group results for mouse weight, TC, HDL cholesterol concentration, and quantification of qPCR, protein relative expression were analyzed by Analysis of Variance (ANOVA) and Tukey’s multiple comparisons test, as appropriate. The values presented are the means ± SD of three independent experiments. Means with different superscript letters are significantly different, *p* < 0.05.

## 3. Results

### 3.1. In Vivo Determine the Dosage and Toxicity of DLW Extracts

Previous studies have evaluated the cytotoxicity of DLW (*Dendropanax morbifera* water) extracts in vitro [6]. Therefore, to verify in vivo test, we were conducted a preliminary experiment to determine the concentration in Figure 1. The low and high concentrations of DLW extract were orally administered for a short or long term to confirm the change of mouse body weight and liver toxicity. We observed changes in body weight and general condition for the toxicity evaluation. After oral administration for 8 days (short-term) and 30 days (long-term), mouse body weight and condition were not affected by DLW extract dose (250 mg/kg, DLW250 and 500 mg/kg, DLW500), compared with the results for the control, dextrin, and *Garcinia* (Figure 1A,B).

Next, to examine the toxicity of DLW extracts, we performed a histological analysis of mouse liver tissue. In both DLW250- and DLW500-treated mice, immunohistochemistry did not show significant signs of liver toxicity. There were no lesions in liver tissue after 8 or 30 days of administration (Figure 1C,D). Additionally, to determine tissue damage and organ toxicity, we measured the weights of whole organs (the liver, kidney, and spleen) by dissection. The weights of the liver, kidney, and spleen of the treatment group were not reduced compared to those of the control group, Dex100, and GAR400. Moreover, when the organs were visually inspected for lesions, there were no signs of damage (Figure 1E,F). Overall, side effects were not observed in normal mice treated with DLW extract (250 mg/kg and 500 mg/kg) for 8 days or 30 days, regarding behavior, organ damage or weight change, and liver pathological examination. Therefore, these results of the preliminary experiment indicate that DLW extracts did not show toxicity and determined the concentration of DLW extract through in vivo toxicity. Moreover, the toxicity and anti-obesity efficacy of DLW extract could be inferred.

### 3.2. Anti-Obesity Function of DLW Extracts In Vivo

Obesity is affected not only by genetic susceptibility but also by the environment. It is very difficult to study the etiology of obesity in humans [12]. Therefore, it is desirable to identify the causes of obesity using animal models.

To evaluate the effects of DLW extracts on body weight and body fat, they were orally administered for 12 weeks to C57BL/6 mice receiving a high-fat diet. The body weights of experimental animals vary day to day. Because obesity is also variable, the body weight was always measured at the same time and in the same environment to minimize this effect. The weight change during the experimental period in the normal control, high-fat (HF) control, HF+Dextrin (100 mg/kg; Dex100), HF+*Garcinia* (400 mg/kg; GAR400), HF+DLW250, and HF+DLW500 groups is shown in Figure 2A. Experimental groups with the same mean body weight started the relevant diet after one week acclimation period.

The HF control group showed a statistically significant difference in mean body weight after 6 weeks compared to the normal control group. This weight gain continued until 12 weeks. After 12 weeks, the mean body weight of the 250 mg∙kg^−1^∙d^−1^ and 500 mg∙kg^−1^∙d^−1^ DLW-treated HF groups was significantly decreased compared to that of the HF control group.

Figure 2B shows the weighed organ tissue for each group. The liver and spleen weights of each group were not statistically different. However, the WAT weight showed a statistically significant difference. Both epididymal WAT and perirenal WAT were assessed. Perirenal and epididymal WAT in the HF+Dex100, HF+GAR400, HF+DLW250, and HF+DLW500 groups were significantly different from the values in the corresponding high-fat groups. These results show that DLW500 inhibits lipid accumulation in mice with obesity induced by a high-fat diet (*p* < 0.05). We also observed the ALT, AST, creatinine, and urea levels in serum to study toxicity and liver damage. DLW decreased the elevated ALT, AST, and creatinine levels in mice with high-fat diet-induced obesity. But, urea level was not affected (Figure 2D). After treatment with the DLW extract, H&E staining was performed from a liver biopsy to the tissue to evaluate liver toxicity and liver fat content. As a result, the HF group was able to observe severe fat accumulation and hepatocellular degeneration compared to the control group. On the other hand, DLW-treated group significantly reduced fat accumulation and hepatocellular degeneration (Figure 2E).

### 3.3. DLW Extract Affects Oxidative Stress and Cholesterol Level in the Tissue, Plasma, Urine

Lipid peroxidation causes cell damage through an oxidative deterioration of cell membrane lipids by a free radical chain reaction mechanism [13]. Thus, we measured MDA to confirm the levels of lipid peroxidation. MDA in the HF group was significantly increased, but the treatment of DLW decreased in the concentration of MDA in liver tissue (Figure 3A). In particular, the concentration of MDA in the HF+DLW500 group was lower than that of the control group and decreased by 47.5% compared to the HF group (*p* < 0.05). It was found that pepper seed powder and Curcuma longa powder, which are known to be effective in anti-obesity, decreased the content of lipid peroxidation in liver tissue in rats fed a high-fat diet [14,15]. Reactive oxygen species (ROS) produced under normal conditions are rapidly eliminated by antioxidant defense mechanisms [16]. Antioxidant enzyme SOD is a major enzymatic defense against free radicals and helps to protect cells against ROS by intercepting free radicals [17]. As shown in Figure 3B, the SOD activity of plasma in this experiment was 3.07 ± 0.17 U/mL in the HF group, only 43.64% compared to 7.04 ± 0.08 U/mL in the control group, and all DLW-treated groups were significantly higher in the HF+DLW250 group, HF+DLW500 group, 1.8 and 2.1 times that of the HF group, respectively (*p* < 0.05). Nitric oxide (NO) is produced from nitric oxide synthase by nitrogen oxide arginine, and verified antioxidant activity by measuring NO elimination activity [18] of NO activity in the HF+DLW treatment group is shown in Figure 3C. The NO activity of the HF+DLW treatment group decreased in a dose-dependent manner, 33.59 ± 0.54 and 30 ± 0.97 at the concentrations of 250 and 500 mg/kg. As a result, NO activity showed lower than the HF group at all concentrations. In addition, we investigated whether HFD could cause systemic endotoxemia. The concentration of plasma endotoxins was higher in the HD group than in the control group (Figure 3D). These results indicate that DLW decreased the plasma endotoxin level.

Cholesterol plays many important roles in vivo, and its function is essential for cell metabolism because it is used to produce and maintain all cell membranes; the total amount of cholesterol is approximately 0.2% of body weight [19]. The blood cholesterol concentration is mainly dependent on the production, absorption, and catabolism of cholesterol in the liver and intestinal tract [20]. The measurement of TC content is important as an index of abnormal lipid metabolism in the body [21]. In addition, as the degree of obesity increased, the TC content increased, and serum TC content was significant for obesity. Thus, after 12 weeks, the serum TC level was significantly lower in the HF+Dex100, HF+GAR400, HF+DLW250, and HF+DLW500 groups than in the HF control group (Figure 3E).

HDL cholesterol, which is synthesized in the liver and small intestine and released into the blood, acts to remove cholesterol from the peripheral tissues to the liver [22]. Increased blood levels can prevent a variety of types of arteriosclerosis. Conversely, a decrease is an arteriosclerosis risk signal that is observed in obesity, hypertension, stress, and smoking [23]. The HDL cholesterol level was measured at 12 weeks, and there was a statistically significant increase in the HF+Dex100, HF+GAR400, HF+DLW250, and HF+DLW500 groups. The HDL cholesterol content in the HF+DLW500 group showed the most dramatic change compared to that of the HF control group (Figure 3F).

### 3.4. DLW Extract Regulates AMPK, ACC-1, CPT1, ATGL, and DGAT1 Expression Levels

To explore the potential anti-obesity mechanisms involved in the effects of DLW treatment in the WAT, the expression of AMPK, ACC-1, CPT1α, CPT1β, ATGL, and DGAT1 was measured in the WAT of HF diet-fed C57BL/6 mice.

To further investigate the mechanisms by which DLW relieves obesity and cholesterol, we determined the expression levels of genes involved in de novo lipogenesis and fatty acid oxidation, including transcription factors and lipid metabolism-related enzymes. In fatty acid catabolism, significant changes in β-oxidation-related gene expression (AMPK and CPT1) were noted in the HF+DLW250 and HF+DLW500 groups compared with the control groups, suggesting that DLW influences TG synthesis and oxidation in the WAT of obese mice. As a result, p-AMPK expression in WAT was evaluated by Western blot analysis. Densitometric analyses for p-AMPK are presented as the relative ratio of each protein to total AMPKα; the expression level of p-AMPK protein expression was significantly decreased in the high-fat diet group compared with the control group, but the levels in the HF+Dex100, HF+GAR400, HF+DLW250, and HF+DLW500 groups were significantly increased compared to those in the HF control group (Figure 4A).

When phosphorylation increases due to AMPK activation, there is a positive metabolic effect on lipid metabolism. In this process, phosphorylation of ACC occurs, which inhibits lipid formation by inactivating ACC [24]. ACC is an important enzyme regulating lipid metabolism in tissues such as the liver and muscle and converts carbonate acetyl-CoA to malonyl-CoA. At this point, malonyl-CoA acts as an inhibitor of carnitine pal-mitoyltransferase-1 (CPT1) present in the outer membrane of the mitochondria [25]. The expression levels of ACC-1 mRNA in the WAT were increased in the HF control group, but in comparison, they were significantly reduced in the HF+DLW250 and HF+DLW500 groups (Figure 4B).

Furthermore, the mRNA expression levels of CPT1α and CPT1β, which are related to fatty acid β-oxidation, are shown in Figure 4B. Therefore, we show that CPT1α/β mRNA expression significantly decreased in the high-fat diet group com-pared to the control group, and significantly increased in the HF+DLW250, and HF+DLW500 groups com-pared to the high-fat diet group. Additionally, we observed the mRNA expression levels of the lipolytic enzymes ATGL and DGAT1 with DLW treatment (Figure 4C).

Adipose triglyceride lipase (ATGL) is mainly expressed in adipose tissue and is an enzyme for triglyceride degradation; it has been reported as a rate-limiting lipolytic enzyme in preadipocyte 3T3-L1 cells [26,27]. Correspondingly, the ATGL mRNA level was significantly increased in the HF+DLW250 and HF+DLW500 groups compared with the HF control group in the adipose tissue. Diacylglycerol acyltransferase 1 (DGAT1) is a chemically inactive form of triglycerides which accounts for most of the fat accumulated in the muscle and liver; it is the last step in triglyceride synthesis. When its activity is inhibited, it blocks the triglyceride synthase catalytic reaction, inhibits fat tracking in adipose tissue, and reduces adipocyte size. In addition, the inhibition of DGAT1 is known to increase energy expenditure and protect against high-fat diet-induced obesity [28]. By increasing the expression of uncoupling protein, energy consumption is increased. Thus, we confirmed that the mRNA expression level of DGAT1 significantly decreased in the HF+DEX100, HF+GAR400, HF+DLW250, and HF+DLW500 groups compared with the HF control group (Figure 4C). Thus, we suggest that DLW is a key compound regulating lipid synthesis, lipid degradation, and fatty acid oxidation in the WAT.

## 4. Discussion

Obesity, caused by an imbalance of energy intake and exercise, is closely related to the incidence of diabetes, lipidemia, and metabolic diseases [29]. Obesity reduces the quality of modern life. Recently, efforts have been made to find anti-obesity foods and ingredients derived from natural substances [30]. However, there is a lack of research on the anti-obesity, cholesterol-lowering effects and mechanism of action of Dendropanax morbifera. Therefore, in this study, the anti-obesity activity of DLW extract was evaluated using C57BL/6 mice with high-fat diet-induced obesity, which are widely used for obesity studies.

Previous studies have confirmed the in vitro toxicity of *D. morbifera* in a previous report [6]. *D. morbifera* did not show toxicity in mouse 3T3-L1 cells and showed anti-obesity and cholesterol-lowing effects in 3T3-L1 cells. We have shown that DLW-treated 3T3-L1 cells did not exhibit cytotoxicity at concentrations of DLW of 0–500 μg/mL over 24 h. In addition, in 3T3-L1 cells treated with DLW for 3 to 9 days, fat accumulation was significantly inhibited, and the numbers of lipid droplets in the cells decreased. Moreover, DLW treatment decreased levels of C/EBPα and FAS, the key regulators for adipogenesis, showing anti-obesity activity due to the inhibition of adipogenesis during differentiation [6]. Therefore, to confirm the in vivo efficacy of *D. morbifera* in mice, the toxicity was first confirmed in Figure 1.

In this study, the effects of DLW extract on weight and body fat in high-fat diet-fed C57BL/6 mice were remarkable, as compared to the mice receiving the control diet. The body weight of the HF+DLW500 group had significantly decreased compared to that of the high-fat diet group. Therefore, dietary intake did not affect the weights of the liver, kidney, or spleen. However, the weights of the epididymal and perirenal WATs of the control group were significantly different compared to the those of the HF diet group. Moreover, H&E staining with liver tissue observed that fat accumulation and hepatocyte degeneration seriously progressed in the HF group compared to the control group, but the DLW-treated group significantly decreased fat accumulation and hepatocellular degeneration compared to the HF group. Thus, these results proposed that DLW extract effectively reduces body weight, WAT weight, fat accumulation, and hepatocyte degeneration.

Lipid peroxidation reaction causes oxidative decomposition by free radicals of polyunsaturated fatty acids in biological tissue membranes, and MDA is used as an indicator, and its increase indicates an increase in oxidative stress (Figure 3A). As a result, this leads to a decrease in the function of the membrane, decreases in fluidity, and impairment of homeostasis, which is a major cause of chronic adult diseases such as diabetes, arteriosclerosis, and cancer and is reported to be particularly related to senescent cells [31]. In addition, it is recognized as the most important mechanism for indicating the degree of tissue damage or physiological phenomena caused by various toxic compounds, drugs, and diseases. This contributes to increasing the oxidative stress of cells in tissues and lowering the antioxidant defense power in vivo [7]. Moreover, SOD is primarily involved in cell defense against oxidative damage caused by superoxide anion radicals by converting the two molecules of superoxide anion into hydrogen peroxide and oxygen molecules [17]. In this experiment, the increase in SOD activity was significantly increased by 1.73, 2.03, 1.81, and 2.1 times in the HF+Dex100, HF+GAR400, HF+DLW250, and HF+DLW500 groups compared to the HF group. Additionally, Nitric oxide (NO) was released from vascular endothelial cells. Its mechanism is synthesized from L-arginine by NO synthase (eNOS). eNOS is calcium-dependent and a useful factor related to blood flow, arterial blood pressure, and vascular tone control. Thus, when HF+DLW was treated, the concentration of NO in the urine of mice was significantly reduced by 73.75 and 65.88%, respectively, compared to the HF group (*p* < 0.05) (Figure 3C).

The lipid concentration in the serum of DLW diet-fed mice was significantly lower than that of the high-fat diet group and the high-fat diet group treated with Dextrin, Garcinia cambogia extract. Furthermore, HDL cholesterol levels in the serum were significantly increased in both the HF+DLW250 and HF+DLW500 groups compared with the high-fat diet group. Because serum cholesterol concentration is mainly influenced by factors related to cholesterol production, absorption, and catabolism in the liver and intestinal tract, TC content is an important indicator of lipid metabolism in the body. The HDL cholesterol released into the blood acts to remove cholesterol from the peripheral tissues to the liver, and an increased serum HDL cholesterol concentration can prevent various types of arteriosclerosis. Thus, the effect of DLW extract on serum lipids could be important in preventing obesity.

We investigated the lipid-regulatory mechanisms in the WAT of obese mice. DLW extract inhibited the expression of lipogenic genes such as AMPK, ACC-1, CPT1α, CPT1β, ATGL, and DGAT1. AMPK plays a major role in maintaining energy homeostasis and is activated through phosphorylation [32]. AMPK also has the effect of increasing lipid oxidation and inhibiting lipid synthesis [33]. As the activity of phosphorylated AMPK increases, it affects the expression of SREBP-1c and inhibits fat synthesis [34]. Therefore, the degree of AMPK protein expression associated with the adipogenic mechanism confirmed in the WAT of C57BL/6 obese mice. In our study, the expression of p-AMPK (β-oxidation-related genes) increased in the HF+Dex100, HF+GAR400, HF+DLW250, and HF+DLW500 groups. SREBP-1c is a transcription factor downstream of AMPK that plays an important role in synthesizing triglycerides in adipose and liver tissues. It regulates the gene expression of enzymes such as acetyl-CoA carboxylase (ACC)-1 which are involved in synthesizing triglycerides in the liver [35]. The function of CPT1 is to catalyze fatty acidization in vivo, and it moves long-chain fatty acids into the mitochondria for β-oxidation. CPT1 plays a major role in the β-oxidation of fats in mitochondria [25]. We confirmed that CPT1α/β increased in the HF+GAR400, and HF+DLW500 groups, respectively. Additionally, fatty acid synthesis-related gene expression (ACC-1) was significantly reduced in the HF+DLW250 and HF+DLW500 groups as compared to the HF control group. Therefore, when phosphorylation of ACC (p-ACC) is promoted by activation of AMPK and ACC enzyme activity is inhibited, the amount of malonyl-CoA is reduced, resulting in fatty acid oxidation. In addition, the amount of ATP increases with the increase in oxidation of fatty acids and contributes to a reduction of body fat [35]. Furthermore, expression levels of the triglyceride degradation- and synthesis-related genes ATGL and DGAT1 were significantly changed in the HF+DLW250 and HF+DLW500 groups and the HF+DEX100, HF+GAR400, HF+DLW250, and HF+DLW500 groups, respectively.

The key point is that DLW extract influenced lipid metabolism functions such as β-oxidation and TG synthesis in the WAT of obese mice.

## 5. Conclusions

In conclusion, DLW extract decreased body weight, body fat, fat accumulation and hepatocellular degeneration in an in vivo animal experiment using HFD-induced obese mice and liver tissue AST, ALT activity, MDA, and plasma SOD of all DLW treatment groups. AST, ALT activity, and MDA concentration decreased compared to the HF group, and SOD activity increased compared to the HF group (*P* < 0.05). In particular, the ALT, AST activity, and MDA concentration of the HF+DLW500 group decreased to the level of the control group. Moreover, NO production was significantly inhibited in various HF+DLW groups. In addition, our results showed beneficial effects on TC, HDL cholesterol, and inhibition of lipogenic genes. The expression of lipogenic genes, β-oxidation-related gene (AMPK, CPT1), and lipolytic enzymes (ATGL) was enhanced, but fatty acid synthesis-related genes (ACC-1, DGAT1) were reduced. Thus, our results suggest that DLW may be effective in suppressing lipid formation in adipocytes to decrease body fat, and DLW is an important extract to control the lipid-regulatory mechanisms that regulates lipid synthesis, lipid degradation, and fatty acid oxidation. Taking these findings into consideration, DLW extract can be applied in the development of functional anti-obesity and cholesterol-lowering foods as an industrially useful material with body fat-reduction activity.

## Figures and Tables

**Figure 1 nutrients-13-01424-f001:**
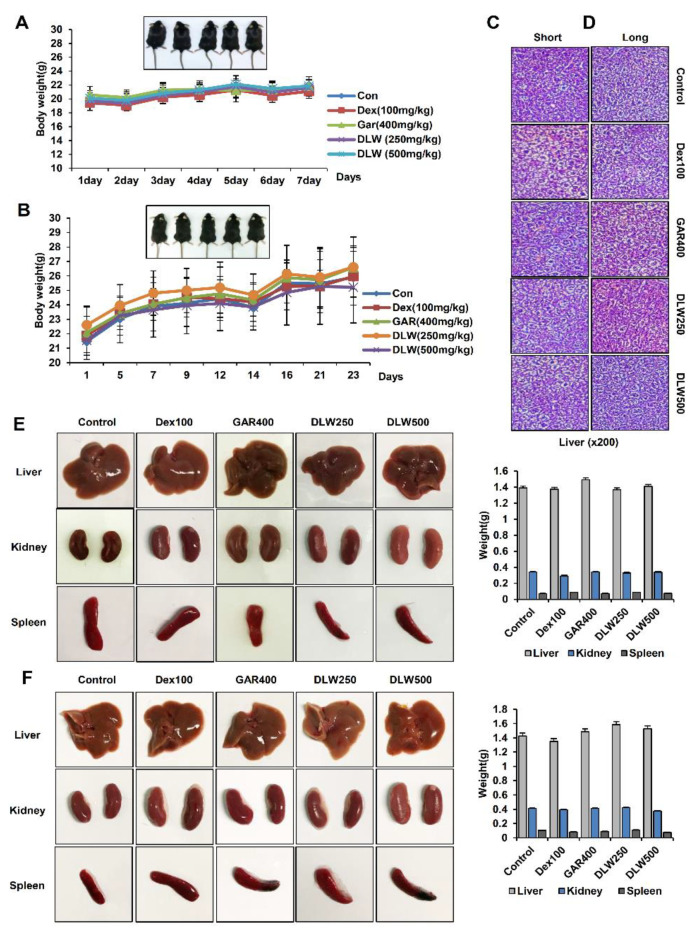
In vivo toxicity of *D. morbifera* water extracts (250 mg/kg, DLW250; 500 mg/kg, DLW500). DLW250 and DLW500 did not demonstrate short-term or long-term toxicity in the in vivo mouse model. (**A**) In vivo short-term toxicity of DLW250 and DLW500. (**B**) In vivo long-term toxicity of DLW250 and DLW500. (**C**,**D**) After 8 or 30 days of DLW250 and DLW500 treatment, hematoxylin and eosin staining of mouse liver specimens was carried out. Representative images are shown. (**E**) In vivo short-term toxicity of DLW250 and DLW500 with respect to organ (the liver, kidney, and spleen) weight. (**F**) In vivo long-term toxicity of DLW250 and DLW500 with respect to organ weight.

**Figure 2 nutrients-13-01424-f002:**
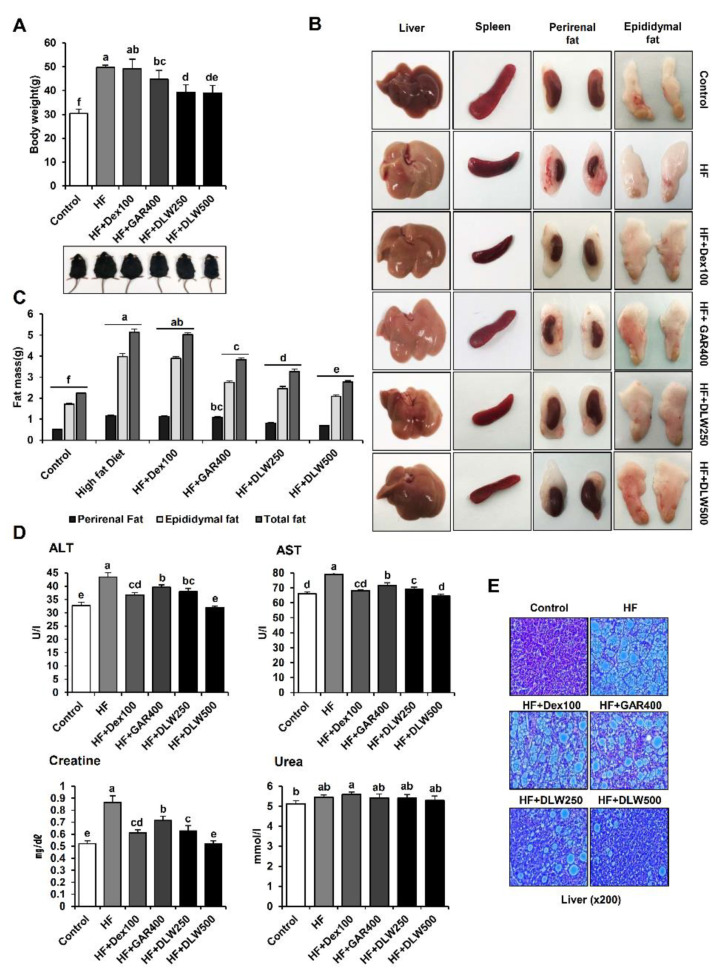
In vivo toxicity of *D. morbifera* water extract in mice with high-fat diet-induced obesity. DLW extract did not show short-term or long-term toxicity in mice. (**A**) Body weight changes in the control, HF (high-fat), HF+Dex100 (high-fat + Dextrin 100 mg/kg), HF+GAR400 (high-fat + *Garcinia* 400 mg/kg), DLW250 (high-fat + DLW 250 mg/kg), and DLW500 (high-fat + DLW 500 mg/kg) groups. (**B**) Representative images of the liver, spleen, epididymal fat, and perirenal fat in each mouse group. (**C**) Measurement of the liver epididymal and perirenal fat mass. The HF, HF+Dex100, HF+GAR400, HF+DLW250, and HF+DLW500 groups were treated for 12 weeks. Values for the liver epididymal and perirenal fat mass are shown as the mean ± SEM (n = 6/group, right panel). (**D**) Toxicity and liver damage in mouse serum. The enzymatic activities (U/L, mg/dL, and nmol/L) are shown for various organs in mice administered DLW (250 and 500 mg/kg body weight) or water as a control. ALT, AST, creatinine, and urea levels were measured in the plasma of mice fed HF, HF+Dex100, HF+GAR400, or HF+DLW (250 and 500 mg/kg body weight). (**E**) After 12 weeks of DLW250 and DLW500 treatment, Histological findings of the liver tissue, hematoxylin and eosin staining of mouse liver specimens was carried out. Representative images are shown. Magnification ×200. Control, fed standard as control; HF, fed high-fat diet; HF+Dex100, fed high-fat diet in conjunction with Dextrin 100 mg/kg; HF+GAR400, fed high-fat diet in conjunction with Garcinia 400 mg/kg; HF+DLW250, fed high-fat diet in conjunction with *D. morbifera* leaf water extracts 250 mg/kg; HF+DLW500, fed high-fat diet in conjunction with *D. morbifera* leaf water extracts 500 mg/kg. The values presented are the means ± SD of three independent experiments. Means with different superscript letters are significantly different, *p* < 0.05 (one-way analysis of variance ANOVA followed by Tukey’s multiple test).

**Figure 3 nutrients-13-01424-f003:**
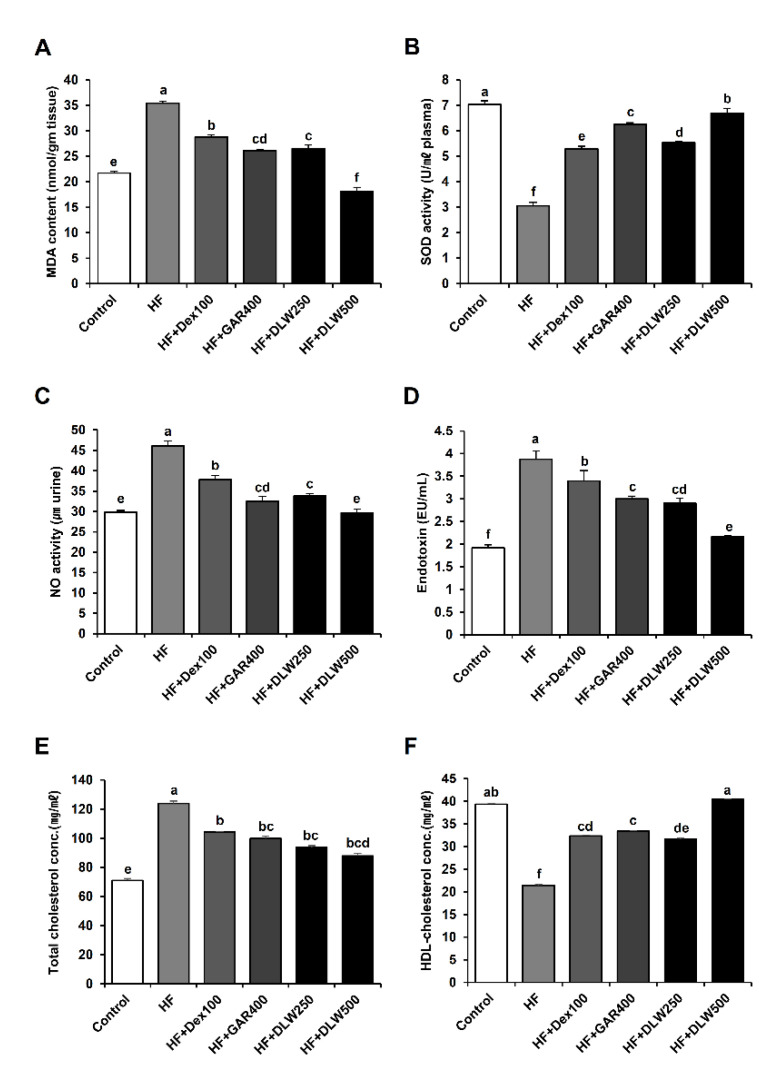
Effects of DLW on relative oxidative stress, cholesterol levels in tissue, serum, urine. MDA and NO activity levels were decreased. SOD activity levels were increased by HF+Dex100, HF+GAR400, HF+DLW250, and HF+DLW500. MDA, SOD, and NO were analyzed in extracts from the liver tissue, plasma, and urine following treatment with HF, HF+Dex100, HF+GAR400, HF+DLW250, and HF+DLW500. Plasma endotoxin concentration (EU/mL) as measured by the LAL assay following treatment with HF, HF+Dex100, HF+GAR400, HF+DLW250, and HF+DLW500. Total cholesterol and HDL cholesterol levels were affected by HF, HF+Dex100, HF+GAR400, HF+DLW250, and HF+DLW500. Total cholesterol and HDL cholesterol were analyzed in extracts from the serum following treatment with HF, HF+Dex100, HF+GAR400, HF+DLW250, and HF+DLW500. (**A**) MDA level. (**B**) SOD activity. (**C**) Nitric Oxide activity. (**D**) Endotoxin level. (**E**) Total cholesterol concentration. (**F**) HDL cholesterol concentration. The values presented are the means ± SD of three independent experiments. Means with different superscript letters are significantly different, *p* < 0.05. (one-way analysis of variance ANOVA followed by Tukey’s multiple test).

**Figure 4 nutrients-13-01424-f004:**
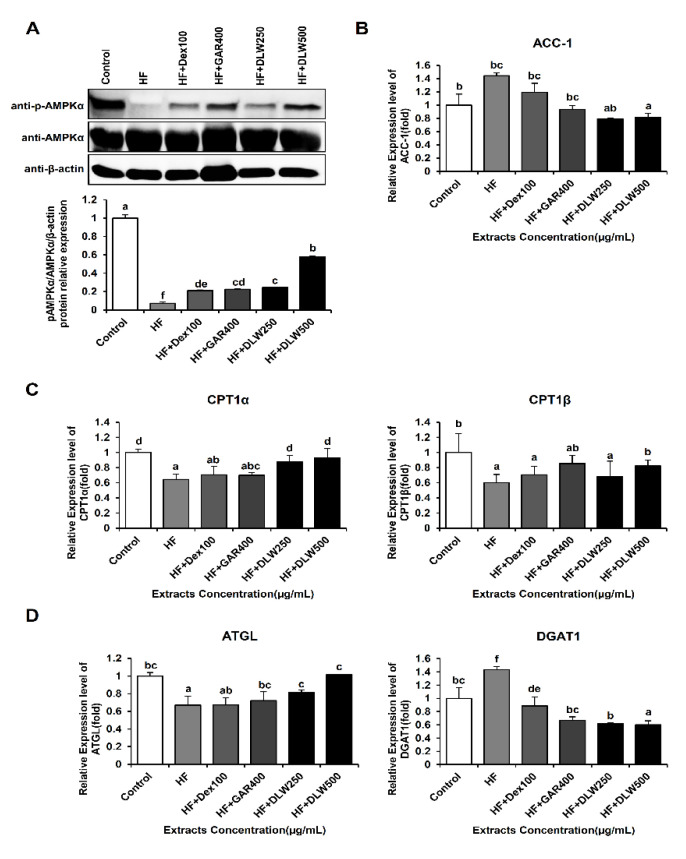
Effect of *D. morbifera* water (DLW) extract on gene expression levels related to lipid synthesis, lipid degradation, fatty acid oxidation, and transcription factors in the white adipose tissue (WAT) of HFD-induced obese mice. DLW influenced protein and mRNA expression of adipogenic genes in mouse WAT. (**A**) After 12 weeks of DLW treatment, the protein expression levels of *p-AMPK* by Western blot analysis were evaluated in WAT from mice fed standard diet, HFD, and HFD with DLW. Densitometric analyses for *p-AMPK* are presented as the relative ratio of each protein to total *AMPKα*, respectively. The mRNA expression levels of (**B**) ACC-1 and (**C**) CPT1α and CPT1β and (**D**) ATGL and DGAT1 by qRT-PCR were evaluated in WAT from mice fed standard diet, HFD, and HFD with DLW. Con, control group fed standard diet; HF, group fed high-fat diet; HF + Dex100, group fed high-fat diet in conjunction with Dextrin 100 mg/kg; HF + GAR400, group fed high-fat diet in conjunction with Garcinia 400 mg/kg; HF + DLW250, group fed high-fat diet in conjunction with *D. morbifera* water (DLW) extract 250 mg/kg; HF + DLW500, group fed high-fat diet in conjunction with *D. morbifera* water (DLW) extract 500 mg/kg; *p-AMPK*, phospho-AMP-activated protein kinase; *AMPK*, AMP-activated protein kinase; *ACC*, acetyl-CoA carboxylase; *CPT1α,* carnitine palmitoyltransferase-1 alpha; *CPT1β,* carnitine palmitoyltransferase-1 beta; *ATGL,* Adipose triglyceride lipase; *DGAT1,* Diacylglycerol acyltransferase 1. The values presented are the means ± SD of three independent experiments. Means with different superscript letters are significantly different, *p* < 0.05. (one-way analysis of variance ANOVA followed by Tukey’s multiple test).

## Data Availability

Not applicable.

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
