# Peer review of "In Vivo Evaluation of Dendropanax morbifera Leaf Extract for Anti-Obesity and Cholesterol-Lowering Activity in Mice"

_nutrients, 2021, doi:10.3390/nu13051424_

Round 1
Reviewer 1 Report
This is a well done and interesting paper, which in vivo evaluates the effect of Dendropanax Morbifera leaf extract for anti-obesity and cholesterol-lowering activity in mice. The study analyzes the fat content of the liver and the serum cholesterol levels in mices trated with D. Morbifera. Interestingly, the Authors also studied the behaviour of oxidant stress in mices. I suggest to relate the MDA and NO activity to endotoxin levels, because the relationship between LPS and oxidant stress has been demonstrated in NAFLD patients. In addition the Authors should analyze serum sp-NOX2 activity and urinary 8-iso.PGF1alpha to complete the study of oxidant stress
Reviewer 2 Report
This manuscript has been updated with more details. However, numerous major issues still remain to be resolved.
Major Compulsory Revisions:
- The authors provide some information about the D. morbifera in the introduction. However, the resource of leaves used in this study is still missing in the “Preparation of D. morbifera leaf extracts”, and there is no information about verification of this herb used in this study.
- Add the resource of white adipose for RNA extraction and qPCR.
- Please explain the light part right above the blots (anti-p-AMPKα) of “HF”, “HF+Dex100”, and “HF+GAR400” in Figure 4A.
Author Response
Please see the attachment.

This manuscript is a resubmission of an earlier submission. The following is a list of the peer review reports and author responses from that submission.
Round 1
Reviewer 1 Report
The Authors should study a subgroup of mices, by evaluating in a liver biopsy before and after the treatment, the fat content in the liver. In addition the oxidant stress has to be studied in mices. by analyzing serum and urinary markers, such as serum sp-NOX2 and urinary 8-iso-PGF-1 alpha. Finally could be of interest to study endotoxin, which Is closely related to the oxidant stress, before and after the D. Morbifera treatment. AST and ALT have been evaluated ?
Reviewer 2 Report
In this study, the authors showed the anti-obesity effects of Dendropanax morbifera leaf extracts, extending their in vitro study and confirming it in vivo. Although this reviewer found the research to be of interest, there are a number of issues that need to be addressed.
Major points
- Lines 14-18: Since this is an international journal, domestic information on obesity in Korea is not necessary in the abstract. Instead the abstract needs to clearly state the purpose of the research.
- Lines 72-74: If these data are to be presented for the toxicity study of Dendropanax morbifera presented in this study, the purpose of the study should be stated more explicitly in the introduction.
- Lines 93-96: If DLW is an aqueous solution (lines 83-84), how was this dosage determined? If it was weighed at the time of freeze-drying and then dissolved in a certain amount of water, this should be stated. Also, what is the basis for determining the dosage?
- Line 140: A multiple group test cannot be performed with a t-test. It is necessary to test all the results using an appropriate test method and reinterpret the results.
- Figure 1: Why didn't the authors administer the drug for 12 weeks, the same period in which they saw the anti-obesity effect, and see if there was any toxicity? If the authors want to ensure safety as a food, 4 weeks is not enough.
- Lines 245-298: This part contains the results and discussion, and the “Discussion” contains the results of gene expression. The interpretation of the results should basically be described in the “Discussion”.
Minor points
- Lines 37-40: Since this sentence describes adipocyte differentiation in vivo, it is not accurate to describe dexamethasone as an obesity factor.
- Lines 54-55: Cardiovascular disease does not include hyperlipidemia and diabetes.
- Line 89: Did the authors use lactating 3-week-old mice?
- Line103: The meaning of ”250 or 500 mg/kg of solvent” is unclear.
- Line 116: It should be creatine, not creatinine.
- Figure 1: Figure 1D is missing.
- Figure 4A: The error bar is missing in ACC-1.
Reviewer 3 Report
This manuscript by Song et al. aims to determine the anti-obesity activity of Dendropanax morbifera leaf extract on high-fat diet-induced obesity mice. The authors found that D. morbifera extract reversed the impact of a high-fat diet on LDL cholesterol, HDL cholesterol, and triglycerides. Treatment with D. morbifera extract also modified the expression of AMPK, ACC-1, CPT1α, CPT1β, ATGL, and DGAT1. However, this study has several serious weaknesses in the experimental design, presentation of results, and conclusions.
Major Compulsory Revisions:
- Provide the resource of leaves and how the herb was characterized.
- The information in “Animal experiment” is very confusing.
- Provide information about the high-fat diet.
- What tissue(s) was used for RNA extraction and qPCR?
- The method of statistical analysis is inappropriate. The data should be analyzed using one-way ANOVA.
- The results are poorly presented. A lot of background information in this section. There is no specific data in the whole section. In Figure 1E, the kidney image of the control group is larger than the rest of the group. In Figure 2, the Y-axis of Figure 2C is unclear. There are no error bars in Figure 4A.
- The mRNA expression of AMPK doesn’t correlate to the activity of this kinase. The author should perform Western blot to determine the activation of AMPK.
- The data from this study do not support the conclusions.
Round 2
Reviewer 1 Report
The Authors performed a series of interesting additional experiments on the fat content in the liver. Also they better explained their data about the oxidant stress in mices. Interestingly, they evaluated, as suggested, AST and ALT before and after D. morbifera. My suggestion is to analyze endotoxin, to complete the paper.
Reviewer 2 Report
Although the revised manuscript by Song et al. shows improvements in certain areas, it still contains the following problems that remain to be corrected.
- Abstract lacks purpose.
- The rationale for the determination of the extract dosage is unclear.
- The experiment in Fig. 1 is not sufficient to analyze the toxicity. It should be called a preliminary study to determine the dosage.
- The meaning is unclear regarding the indication of statistical significance.
Reviewer 3 Report
This manuscript has been updated with more details. However, numerous major issues still remain to be resolved.
Major Compulsory Revisions:
- The authors provide some information about the D. morbifera in the introduction. However, the resource of leaves used in this study is still missing, and there is no information about the verification of this herb used in this study.
- What does VHFD stand for? The kcal% and g% are not units used in scientific writing.
- Which part of the white adipose tissues was used for RNA extraction and qPCR?
- The way how the authors performed the statistical analysis is confusing. Please provide the original statistical analysis for review.
- In Figure 4A, the blots of αAMPK were over-exposed which could you affect the results. The label of “αp-AMPK” should be “p-α-AMPK”. Please provide the original Western blot image of p-α-AMPK for review.
